# Exploring Dependence Relationships between Bitcoin and Commodity Returns: An Assessment Using the Gerber Cross-Correlation

Kokulo K. Lawuobahsumo [1,†] , Bernardina Algieri [1,2,†] , Leonardo Iania [3,4,†] and Arturo Leccadito [1,*,†]

1. Department of Economics, Statistics and Finance, University of Calabria, Ponte Bucci, 87030 Rende, Italy
2. Department of Economic and Technological Change, Zentrum für Entwicklungsforschung (ZEF), Universität Bonn, Walter-Flex-Straße 3, 53113 Bonn, Germany
3. CORE/LFIN, Université catholique de Louvain (UCLouvain), Voie du Roman Pays 34, B-1348 Louvain-la-Neuve, Belgium
4. Department Accounting, Finance and Insurance, University of Leuven (KU Leuven), Naamsestraat 69, 3001 Leuven, Belgium
* Correspondence: arturo.leccadito@unical.it
† These authors contributed equally to this work.

**Abstract:** We use a robust measure of non-linear dependence, the Gerber cross-correlation statistic, to study the cross-dependence between the returns on Bitcoin and a set of commodities, namely wheat, gold, platinum and crude oil WTI. The Gerber statistic enables us to obtain a more robust co-movement measure since it is neither affected by extremely large nor small movements that characterise financial time series; thus, it strips out noise from the data and allows us to capture effective co-movements between series when the movements are "substantial". Focusing on the period 2014–2022, we construct the bootstrapped confidence intervals for the Gerber statistic and test the null that all the Gerber cross-correlations up to lag $k_{max}$ are zero. Our results indicate a low degree of dependence between Bitcoin and commodities prices, both when we consider contemporaneous correlation and when we employ correlations between current Bitcoin and lagged (one day, one week, or one month) commodities returns. Further, the cross-correlation between Bitcoin and commodities' returns, although scanty, shows an increasing trend during periods of economic, health and financial turbulence. This increased cross-correlation of returns during hectic market periods could be due to the contagion effect of some markets by others, which could also explain the strong dependence across volatilities we detected. Based on our results, Bitcoin cannot be considered the "new digital gold".

**Keywords:** Gerber correlation; cross-correlation; comovements; Bitcoin

## 1. Introduction

Cryptocurrencies, defined as decentralised digital currencies that rely on encrypting to verify transactions, have gained increased popularity among retail and institutional investors as a new interesting asset class. Launched in 2009, the total market capitalisation of crypto-assets has experienced exponential growth, passing from about EUR 20billion in January 2017 to more than EUR 3 trillion by the end of 2021 and decelerating in the second quarter of 2022. In this context, the nature of dependence across returns of different asset classes becomes essential for portfolio allocation, policy formulation and asset pricing.

The present study investigates the cross-dependence between the returns of cryptocurrencies and commodities in order to determine how strongly the assets are interlinked and how they can influence each other. We focus on the most renowned cryptocurrency, Bitcoin, and compute robust measures of non-linear dependence between Bitcoin's returns and the leads/lags returns of four different commodities. These latter belong to the categories of precious metals (platinum and gold), energy (crude oil WTI) (West Texas Intermediate (WTI) crude oil is a light, sweet, high-quality crude oil sourced primarily from inland Texas

that serves as a global benchmark in oil pricing along with the European BRENT extracted in the North Sea) and agricultural products (wheat).

Previous studies investigating the relationship and volatility spillover (see [1] for a comprehensive literature overview) between Bitcoin and gold/oil prices (or returns) have produced mixed findings. Ref. [2] explores the linkages between Bitcoin and gold prices while [3] investigates the connection between (i) Bitcoin and (ii) crude oil and gold prices. Both studies conclude that the dynamics of gold (and oil) prices do not have a significant impact on cryptocurrencies' returns. Using transfer entropy, Ref. [4] investigates the link between gold and cryptocurrency prices and shows that gold could be a good hedging instrument for cryptocurrencies. These results are in line with the findings of [5], who evaluates the time-varying conditional correlations between Bitcoin and gold returns using the BEKK-GARCH model. The author shows that Bitcoin and stock market returns are positively correlated during financial market downturns, in sharp contrast to the behaviour of gold returns, which is widely believed to be a hedging instrument against stock market downfalls. These findings are challenged by (i) [6,7], who show that gold is very sensitive to uncertainty shock from cryptocurrency markets, and by (ii) [7], who employs a time-varying parameter vector autoregressive model to show that gold is vulnerable to return and volatility spillovers from cryptocurrency uncertainty measures. The difference in behaviour between Bitcoin and commodities returns is carried out also for higher-order moments. Ref. [8] reports a significant difference in the long-term volatility of Bitcoin compared to gold returns. Ref. [8] attribute this result to the fact that since Bitcoin does not have an income stream or an intrinsic value, its price tends to be more sensible non-fundamental financial markets news/sentiment. Finally, Ref. [9] assesses the impact of global economic policy uncertainty and natural resource prices (oil and gold in particular) on Bitcoin returns. By employing an autoregressive distributed lagged (ARDL) model and a nonlinear ARDL model for evaluating the symmetric and asymmetric long- and short-run effects, Ref. [9] observes that oil price had a negative relationship with Bitcoin. Furthermore, using a partial sum of positive and negative changes in global policy uncertainty, gold price, and oil price as the asymmetric long-run equation of Bitcoin return, Ref. [9] reports that (i) asymmetry shocks in oil price positively impact Bitcoin returns and that (ii) a positive (negative) shock in the gold price is negatively (positively) related to Bitcoin returns.

We use the Gerber statistic as a tool to capture the dependence between the time series of Bitcoin log returns and (leads/lags) of the time series of commodities log returns. The measure we employ was introduced by [10], and it is a robust measure of pairwise movements of two series of returns. In particular, it counts the proportion of co-movements in the series of interest, i.e., when both series simultaneous pierce the thresholds specified by the econometrician. As such, it represents an extension of Kendall's Tau. The Gerber statistics has two key advantages with respect to other measures of dependence. First, since only joint co-movements larger than the chosen thresholds enter the statistic, the Gerber correlation is insensitive to small movements in the series that may simply be noise. Second, in contrast to product-moment-based measures, such as the Pearson correlation, the Gerber statistic is insensitive to extreme movements because it relies on the number of times the returns jointly exceed the thresholds and not on the extent to which the thresholds are pierced, Ref. [11] introduces a time-varying version of the Gerber statistic, which is used with the aim of identifying co-movements in commodity prices over the period 2006–2020. Another application of the Gerber correlation to commodity markets is represented by [12]. The authors use rolling windows estimations for Gerber correlation using monthly data over 170 years. Similar to [11] and contrary to [12], where spot commodity prices are employed, we use daily data for futures prices of each of the four commodities; Ref. [11], however, is concerned only with the Gerber correlation for contemporaneous values of each pair of commodities, whereas here we consider what we call the Gerber cross-correlation.

The paper makes the following contributions. It is the first study using Gerber correlation to analyse the dependence between Bitcoin and commodities. Secondly, it considers a cross-correlation version of the measure, i.e., the Gerber statistic between the cryptocurrency and leads or lags log price changes of four commodities of interest. Finally, it relies on the bootstrap to derive confidence intervals for the newly introduced measure and to test the null that all the Gerber cross-correlations up to lag $k_{\max}$ are zero.

The remainder of the study is organised as follows. Section 2 presents the Gerber correlation and cross-correlation statistics and inference methods for the two measures. Section 3 shows the results of the empirical application involving log returns. Section 4 concludes.

## 2. Materials and Methods

### 2.1. The Gerber Statistic

Let $\{(y_{1,t}, y_{2,t}) : t \in \mathbb{Z}\}$ be a strictly stationary bivariate time series. The Gerber statistic is defined as

$$\frac{\mathbb{E}\left[I_{1,t}^U I_{2,t}^U + I_{1,t}^D I_{2,t}^D\right] - \mathbb{E}\left[I_{1,t}^U I_{2,t}^D + I_{1,t}^D I_{2,t}^U\right]}{\mathbb{E}\left[I_{1,t}^U I_{2,t}^U + I_{1,t}^D I_{2,t}^D\right] + \mathbb{E}\left[I_{1,t}^U I_{2,t}^D + I_{1,t}^D I_{2,t}^U\right]} \tag{1}$$

where for $i = 1, 2$

$$I_{i,t}^U = I(y_{i,t} \geq H_{i,t}) = \begin{cases} 1 & \text{if } y_{i,t} \geq H_{i,t} \\ 0 & \text{if } y_{i,t} < H_{i,t} \end{cases}$$

$$I_{i,t}^D = I(y_{i,t} \leq -H_{i,t}) = \begin{cases} 1 & \text{if } y_{i,t} \leq -H_{i,t} \\ 0 & \text{if } y_{i,t} > -H_{i,t} \end{cases},$$

$H_{1,t}$ and $H_{2,t}$ are the thresholds for the two series, and $I(A)$ denotes the indicator function for the event $A$. The sample counterpart of (1) based on $T$ observations is

$$\widehat{g}(0) = \frac{f_0^c - f_0^d}{f_0^c + f_0^d} \tag{2}$$

where

$$f_0^c = \frac{1}{T}\sum_{t=1}^{T} I(y_{1,t} \geq H_{1,t})I(y_{2,t} \geq H_{2,t}) + \frac{1}{T}\sum_{t=1}^{T} I(y_{1,t} \leq -H_{1,t})I(y_{2,t} \leq -H_{2,t}) = f^{UU} + f^{DD}$$

$$f_0^d = \frac{1}{T}\sum_{t=1}^{T} I(y_{1,t} \geq H_{1,t})I(y_{2,t} \leq -H_{2,t}) + \frac{1}{T}\sum_{t=1}^{T} I(y_{1,t} \leq -H_{1,t})I(y_{2,t} \geq H_{2,t}) = f^{UD} + f^{DU}.$$

Hence, $f_0^c$ denotes the proportion of concordant pairs, i.e., the number of times both series pierce their thresholds while moving in the same direction divided by $T$. Indeed, $f_0^c$ is equal to the sum of $f^{UU}$, the proportion of pairs in the sample for which both series are larger than their threshold, and $f^{DD}$, the proportion of pairs for which both $y_1$ and $y_2$ are smaller than their threshold times minus one. On the other hand, $f_0^d = f^{UD} + f^{DU}$ represents the frequency of discordant pairs in the sample, i.e., the number of times both series pierce their thresholds while moving in the opposite direction divided by $T$. Note that the statistic in (2) coincides with Kendall's Tau if the thresholds $H_{1,t}$ and $H_{2,t}$ are equal to zero for all $t$.

### 2.2. The Gerber Cross-Correlation

In this study, we examine the Gerber statistic between $y_{1,t}$ and $y_{2,t-k}$, which we define as Gerber cross-correlation. The sample Gerber cross-correlation is hence defined by the making some straightforward changes to (2):

$$\widehat{g}(k) = \frac{f_k^c - f_k^d}{f_k^c + f_k^d} \tag{3}$$

where for $k = 0, \pm 1, \pm 2, \ldots$

$$f_k^c = \frac{1}{T-k} \sum_{t=k+1}^{T} [I(y_{1,t} \geq H_{1,t})I(y_{2,t-k} \geq H_{2,t-k}) + I(y_{1,t} \leq -H_{1,t})I(y_{2,t-k} \leq -H_{2,t-k})]$$

$$f_k^d = \frac{1}{T-k} \sum_{t=k+1}^{T} [I(y_{1,t} \geq H_{1,t})I(y_{2,t-k} \leq -H_{2,t-k}) + I(y_{1,t} \leq -H_{1,t})I(y_{2,t-k} \geq H_{2,t-k})].$$

The Gerber cross-correlation is a non-linear pairwise dependence measure counting simultaneously the number of piercings of the thresholds. Similarly to the Pearson's correlation coefficient (and Kendall's Tau coefficient), the Gerber statistics lies in the interval $[-1, 1]$. However, contrarily to the Pearson's statistics, the Gerber correlation coefficient does not include all available the data points in its computation. In contrast, it is computed by including only meaningful (i.e., above the threshold) co-movement of the pairs of returns, thus being more robust to small positive/negative co-movements.

Inference Methods

We use the stationary bootstrap of [13] to obtain the confidence intervals for the Gerber cross-correlation and to test its significance. This method consists in a block bootstrap where blocks have random lengths. In particular, we assume that the block length has a geometric distribution. The optimal (average) block length for the stationary bootstrap is selected based on the criterion discussed by [14]. The adopted resampling scheme is needed to preserve the time-series dependence between variables $y_1$ and $y_2$. Gerber cross-correlation based on the stationary bootstrap resample is defined as follows:

$$\widehat{g}^*(k) = \frac{f_k^{c*} - f_k^{d*}}{f_k^{c*} + f_k^{d*}} \tag{4}$$

with

$$f_k^{c*} = \frac{1}{T-k} \sum_{t=k+1}^{T} \left[ I(y_{1,t}^* \geq H_{1,t}^*)I\left(y_{2,t-k}^* \geq H_{2,t-k}^*\right) + I(y_{1,t}^* \leq -H_{1,t}^*)I\left(y_{2,t-k}^* \leq -H_{2,t-k}^*\right) \right]$$

$$f_k^{d*} = \frac{1}{T-k} \sum_{t=k+1}^{T} \left[ I(y_{1,t}^* \geq H_{1,t}^*)I\left(y_{2,t-k}^* \leq -H_{2,t-k}^*\right) + I(y_{1,t}^* \leq -H_{1,t}^*)I\left(y_{2,t-k}^* \geq H_{2,t-k}^*\right) \right].$$

where $(y_{1,t}^*, y_{2,t}^*)$ is the bootstrap sample and $H_1^*$ and $H_2^*$ are the thresholds based on the bootstrap sample. To test the null $H_0 : g(1) = g(2) = \ldots = g(k_{\max}) = 0$ we consider the Box–Pierce test statistic

$$\widehat{Q}(k_{\max}) = T \sum_{k=1}^{k_{\max}} \widehat{g}^2(k)$$

and construct $B$-centred bootstrap realisations of $\widehat{Q}(k_{\max})$, namely

$$\widehat{Q}^*(k_{\max}) = T \sum_{k=1}^{k_{\max}} [\widehat{g}^*(k) - \widehat{g}(k)]^2.$$

Iterating the stationary bootstrap procedure $B$ times, we end up with $B$ sets of Gerber cross-correlation, $\left[\widehat{g}_b^*(1),\dots,\widehat{g}_b^*(k_{\max})\right]_{b=1}^B$, and the corresponding test statistics

$$\left[\widehat{Q}_b^*(k_{\max}) = T\sum_{k=1}^{k_{\max}}[\widehat{g}_b^*(k) - \widehat{g}(k)]^2\right]_{b=1}^B.$$

To carry out the test, we compute the bootstrap $p$-value as

$$\widehat{p}(k_{\max}) = \frac{1}{B}\sum_{b=1}^B I\left[\widehat{Q}_b^*(k_{\max}) \geq \widehat{Q}(k_{\max})\right].$$

### 3. Data and Empirical Results

*3.1. Data Description*

We compute the daily log returns, i.e., changes in log prices, for Bitcoin and the futures prices of the commodities listed in Table 1, where the Bitcoin and crude oil series are labelled BTC and WTI, respectively. Commodities prices are based on the first generic futures contracts series extracted from Bloomberg. We focus our attention on commodity futures prices since they are the sources of many forward-looking decisions of economic agents and represent important price signals to guide future spot prices [15,16]. Indeed, futures prices account for the expectations of supply and demand of the selected commodities. For instance, producers may define their supply strategy based on the price of futures contracts, and investors could outline their asset allocation strategy based on the trend of futures prices. With future prices, we can hence capture market sentiments regarding the commodities since a future contract obligates the seller and a potential buyer to transact at a specified future date and an agreed-upon price. In addition, futures contracts are widespread speculation vehicles, so the link with cryptocurrency markets is more straightforward. The daily prices for Bitcoin come from Yahoo finance. Data cover the period from 18 September 2014 to 17 June 2022 for a total of 1954 observations per series. Table 2 reports the descriptive statistics for the five series of returns.

**Table 1.** Commodity futures (Bloomberg Tickers).

| Selected Commodities | |
|---|---|
| **Ticker** | **Description** |
| CL1 Comdty | Generic 1st Crude Oil WTI Futures |
| PL1 Comdty | Generic 1st Platinum futures |
| W1 Comdty | Generic 1st Wheat futures |
| GC1 Comdty | Generic 1st Gold futures |

Daily returns are not normally distributed, volatile, leptokurtic and, except for wheat, negatively skewed. It can be noticed that excluding platinum, all commodities and Bitcoin provide positive average daily returns. More specifically, mean daily returns range from 0.01% (WTI) to 0.15% (Bitcoin). High volatility makes all average returns not statistically different from zero, even though the median daily returns on Bitcoin and crude oil WTI investments are, in absolute value, up to 9 times higher than the median returns in other commodities. However, these higher returns are characterised by higher volatility, of roughly 4% and 3% for Bitcoin and WTI compared to 1 to 2% for the other commodities, and stronger negative skewness, which would suggest frequent small gains and a few extreme losses; see [17]. Bitcoin's high volatility exposes thus investors to relevant risks that can lead to significant profits or sharp losses.

**Table 2.** Descriptive statistics for daily log-returns of Bitcoin and commodities futures.

|  | BTC | WTI | Platinum | Wheat | Gold |
|---|---|---|---|---|---|
| Mean | 0.0015 | 0.0001 | −0.0002 | 0.0004 | 0.0002 |
| Standard Deviation | 0.0422 | 0.0325 | 0.0168 | 0.0193 | 0.0093 |
| Median | 0.0019 | 0.0013 | 0.0002 | −0.0002 | 0.0003 |
| Minimum | −0.4647 | −0.3454 | −0.1231 | −0.1130 | −0.0511 |
| Maximum | 0.2252 | 0.3196 | 0.1118 | 0.1970 | 0.0577 |
| Standard Error | 0.0010 | 0.0007 | 0.0004 | 0.0004 | 0.0002 |
| Skewness | −0.8523 | −0.7362 | −0.2743 | 0.5836 | −0.0783 |
| Kurtosis | 14.0416 | 29.3225 | 7.9913 | 10.2708 | 7.2735 |
| JB Stat | 10162.5935 | 56588.1481 | 2052.8205 | 4414.9624 | 1488.8657 |
| JB pval | 0.0000 | 0.0000 | 0.0000 | 0.0000 | 0.0000 |

Note: 'JB stat.' and 'JB pval.' denote the Jarque–Bera test statistic and *p*-value, respectively.

Each panel of Figure 1 reports the full-sample Gerber statistics for leads and lags, $k = -25, \ldots, +25$ days, between the Bitcoin returns, $y_{1,t}$ in Equation (3), and $y_{2,t-k}$, the second variable of Equation (3), which is the crude oil WTI (top-left), platinum (top-right), wheat (bottom-left) and gold (bottom-right), respectively. Since for negative values of $k$ for the pair $(y_{1,t}, y_{2,t-k})$ are equivalent to the pair $(y_{2,t}, y_{1,t-j})$, with $j = -k$, Figure 1 provides the whole picture of the cross dependence of Bitcoin (commodities) returns on lagged commodities (Bitcoin) returns. There are two main messages arising from Figure 1. First, independent of the lags/leads or commodity type, the cross-correlation with Bitcoin returns are mild, mostly below 0.1 in absolute value. This result would signal that Bitcoin is widely different from commodity markets, in line with the analysis by [5,18]. Second, as the lags/leads change, correlations tend to switch sign with no clear pattern. In the next section, we assess if these cross-correlations are time varying via a rolling window analysis.

*3.2. Rolling Window Estimation of Gerber Cross-Correlations*

In this section, we use a rolling window procedure to obtain time-varying estimates of the statistic (3). We set (i) the window size to roughly three years, corresponding to 750 daily observations, (ii) $k$ to −25, −5, −1, 0, 1, 5, and 25, corresponding to leads/lags of one day, one week and (roughly) one month, and (iii) the thresholds to one-half of the unconditional volatilities of the original series. For the block bootstrap procedure, we perform 1000 replications by re-sampling blocks of data, instead of individual values, to preserve the cross-sectional dependence of the original series. The time-varying cross-correlations are reported in Figures 2–9. Focusing on the contemporaneous cross-correlation, the top-left panel of each figure, one notices that the cross-correlations have turned (significantly) positive for WTI and precious metals after the COVID-19 crisis. This indicates that the health and economic-financial turmoil increases the connectedness between Bitcoin and the other asset classes—in our case, commodities. Indeed, contagion effects across markets can trigger the increased connectedness of returns during hectic market periods, and this would have a major impact on risk and investment portfolio management, and policy making.

Turning to lags/leads correlations, Figure 2 reports a small, positive correlation between Bitcoin at time *t* and WTI one week or one month lagged. Conversely, the correlation is negative when WTI is one day lagged, except for the period of pandemics, when the cross-correlation becomes positive.

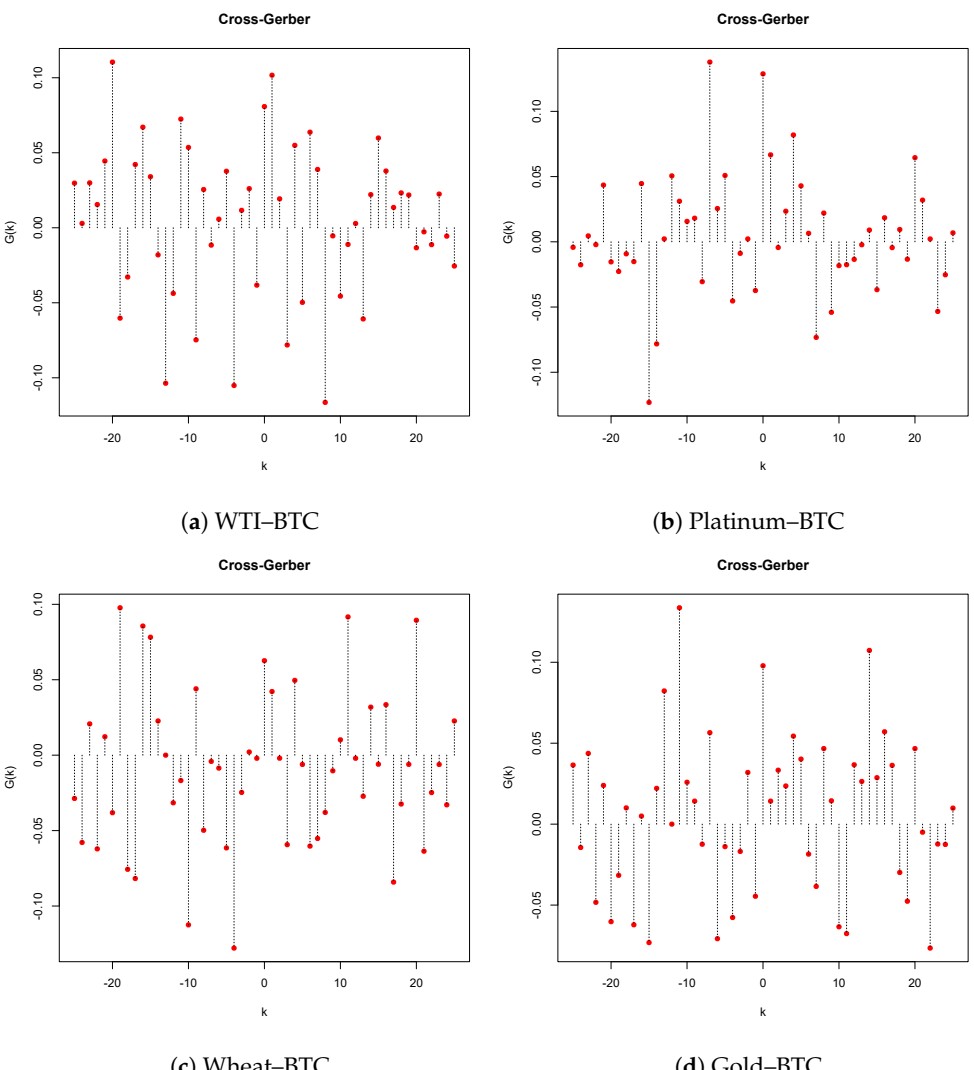

**(a)** WTI–BTC

**(b)** Platinum–BTC

**(c)** Wheat–BTC

**(d)** Gold–BTC

**Figure 1.** Whole Sample Gerber cross-correlations between Bitcoin (BTC) and each commodity. Note: Thresholds are $H_i = \sigma_i/2$ where $\sigma_i$ are the (unconditional) return volatilities for $i = 1, 2$.

When the correlation is computed between WTI at time $t$ and BTC one day lagged, we observe a low but positive correlation (Figure 3b) before the COVID-19 crisis, contrary to when WTI is one day lagged (Figure 2b). In Figure 3c, we detect a higher correlation (around 0.1) between the WTI at time $t$ and Bitcoin at time $t - 5$, compared to a very low correlation (around 0) when instead WTI is one week lagged (Figure 2c). In both cases, we can also notice a sharp change in the correlation during the COVID-19 pandemic.

The contemporaneous Gerber correlations between Bitcoin and platinum's returns are generally positive but low. The same positive nexus exists when Bitcoin is evaluated at time $t$, and platinum's returns are one week or month lagged (Figure 4). When instead the Bitcoin's returns are lagged, the situation is almost similar to WTI (Figure 5). In both cases, we see a generally increasing trend after the Coronavirus pandemics. The cross-correlation between contemporaneous Bitcoin and wheat is positive and declining between 2017 and 2020. It started rising during the pandemic, but overall it remains low and consistently below 0.15 (Figures 6 and 7). Bitcoin and gold display a contained and, only at times, negative Gerber correlation (Figures 8 and 9). This latter result, coupled with the fact that Bitcoin returns' volatility is five times higher than that of gold, highlights the fundamental differences between the two assets. First, while gold is considered a safe haven in times of financial or political uncertainty, the same cannot be said for Bitcoin. Given the low/contrasting correlations between the two assets, investors do not see the

two assets as a substitute for each other. Second, given the high(er) volatility of Bitcoin returns, this asset cannot be considered an investment to rely on in turbulent times, as is the case for gold.

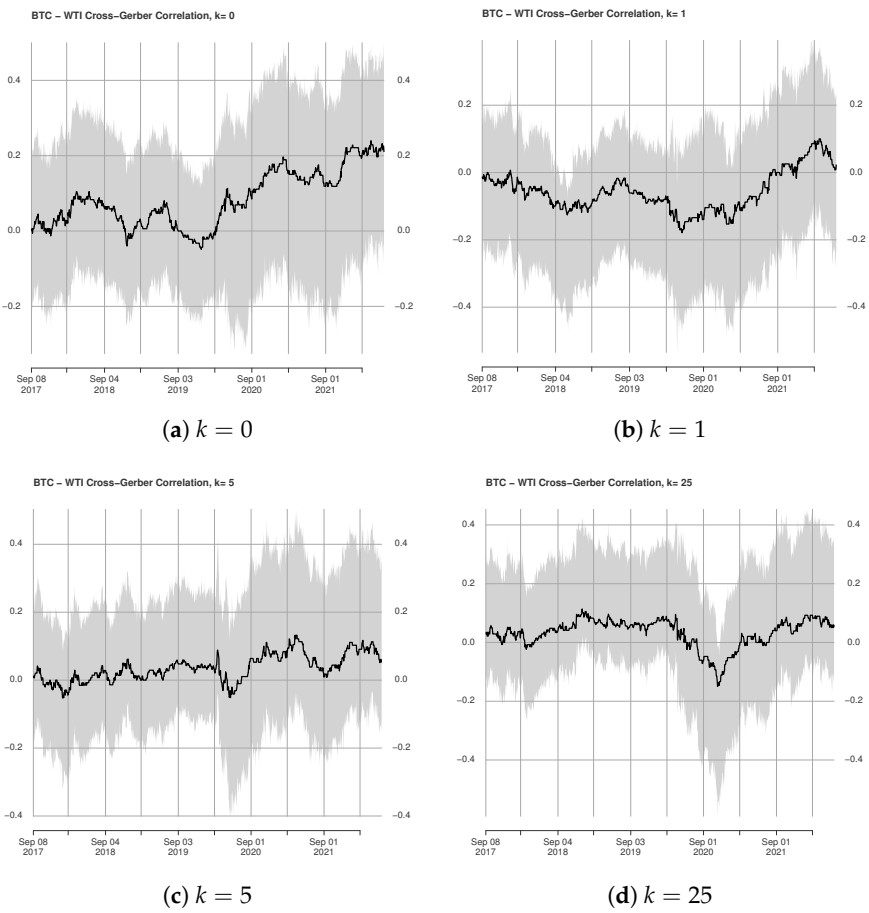

**Figure 2.** The 3-year trailing Gerber cross-correlations and 99% confidence bands, $y_1$ = BTC, $y_2$ = WTI. Note: Thresholds are $H_i = \sigma_i/2$ where $\sigma_i$ are the (unconditional) return volatilities for $i = 1, 2$.

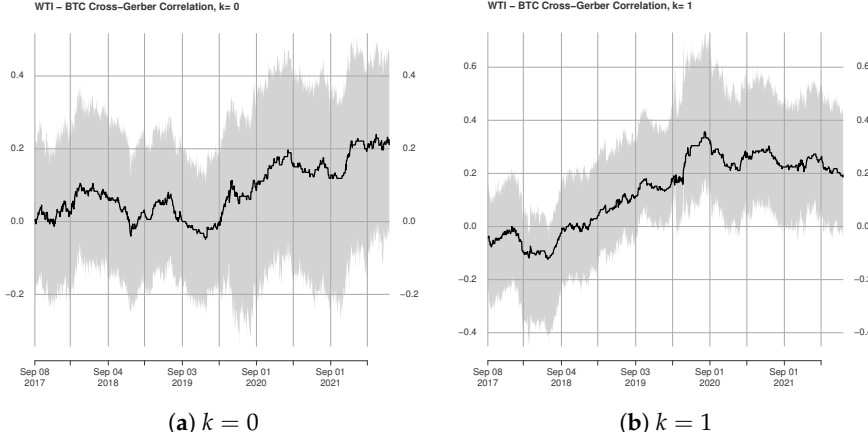

**Figure 3.** *Cont.*

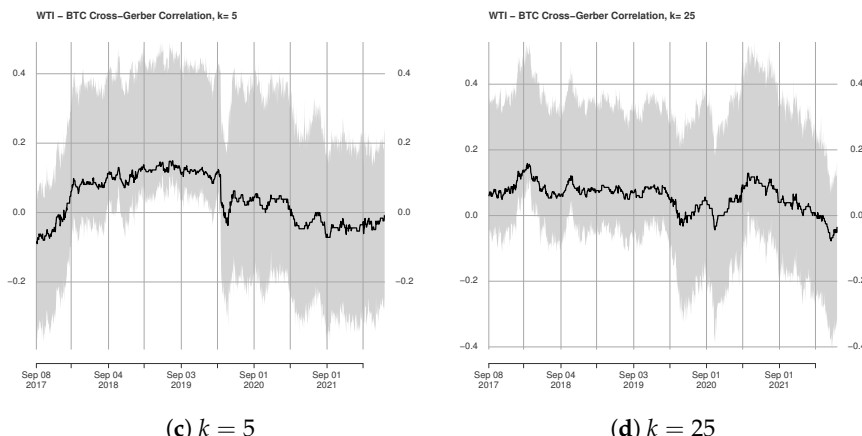

**(c)** $k = 5$        **(d)** $k = 25$

**Figure 3.** The 3-year trailing Gerber cross-correlations and 99% confidence bands, $y_1$ = WTI, $y_2$ = BTC. Note: Thresholds are $H_i = \sigma_i/2$ where $\sigma_i$ are the (unconditional) return volatilities for $i = 1, 2$.

These considerations support the findings by [19] that pointed out how Bitcoin cannot be considered the "new digital gold", i.e., Bitcoin cannot provide a store of value similar to gold nor can it be considered a safe haven like the yellow metal. Our results differ from the study by [20], who documented a negative relation between gold price and Bitcoin. Our findings also diverge from [21], which detected a significant positive connectedness between Bitcoin and gold. The positive connectedness would suggest that Bitcoin has a potential hedging ability like gold, which is opposite to our results.

Our analysis indicates that the cross-correlation between Bitcoin's and commodities' returns is relatively scanty, even if it has increased during the recent turbulent periods. The rising interdependence between commodities and cryptocurrencies in periods of economic crisis is similar to the study by [21] that applied a nonlinear ARDL model to crude oil and Bitcoin. We can attribute our finding to the highly speculative activities of investors during periods of economic, financial, and health turmoil and to capital movements from more risky investments to safe havens. Indeed, these factors may result in high interactions between different markets (see [22]).

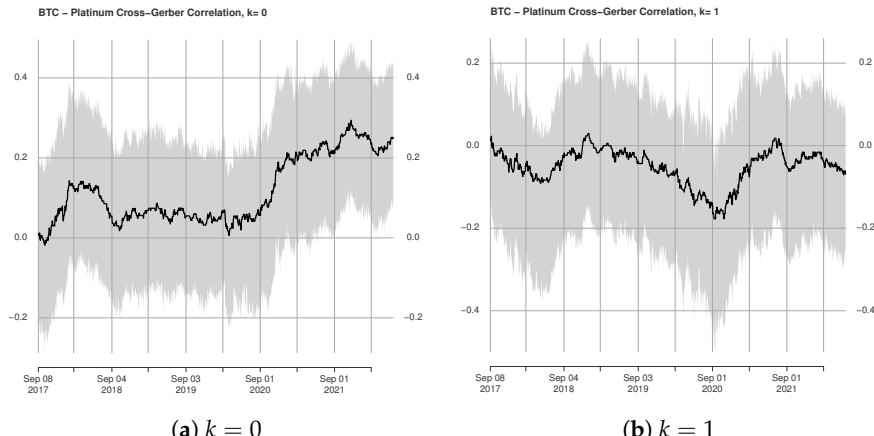

**(a)** $k = 0$        **(b)** $k = 1$

**Figure 4.** *Cont.*

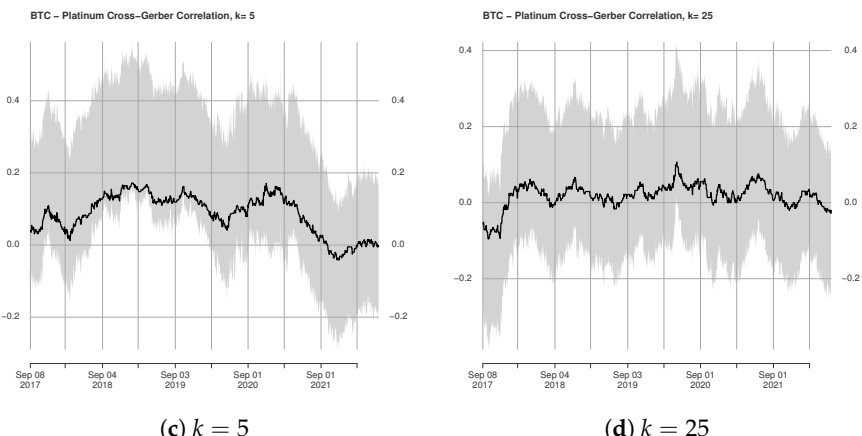

**Figure 4.** The 3-year trailing Gerber cross-correlations and 99% confidence bands, $y_1$ = BTC, $y_2$ = Platinum. Note: Thresholds are $H_i = \sigma_i/2$ where $\sigma_i$ are the (unconditional) return volatilities for $i = 1, 2$.

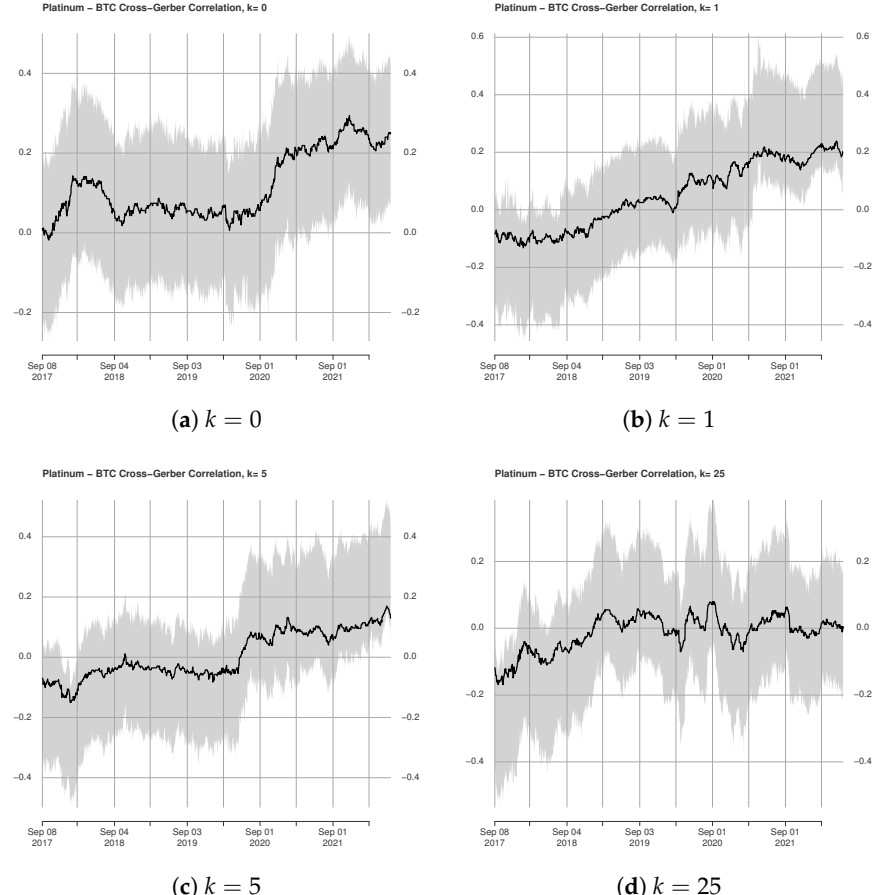

**Figure 5.** The 3-year trailing Gerber cross-correlations and 99% confidence bands, $y_1$ = Platinum, $y_2$ = BTC. Note: Thresholds are $H_i = \sigma_i/2$ where $\sigma_i$ are the (unconditional) return volatilities for $i = 1, 2$.

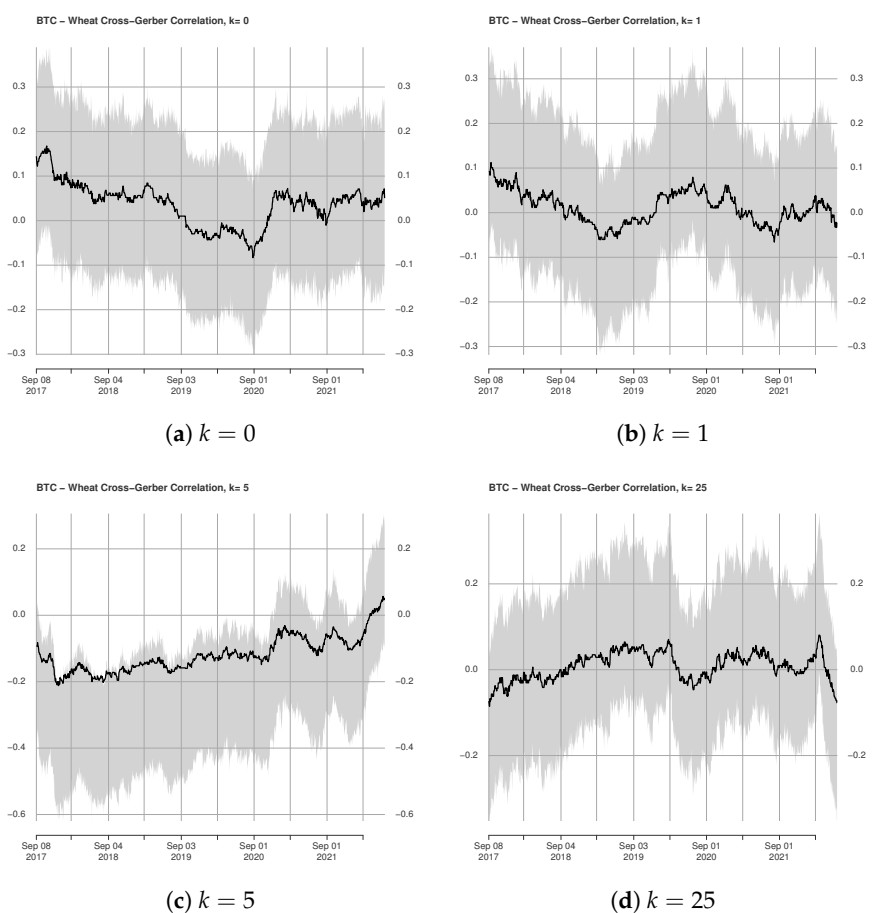

**Figure 6.** The 3-year trailing Gerber cross-correlations and 99% confidence bands, $y_1$ = BTC, $y_2$ = Wheat. Note: Thresholds are $H_i = \sigma_i/2$ where $\sigma_i$ are the (unconditional) return volatilities for $i = 1, 2$.

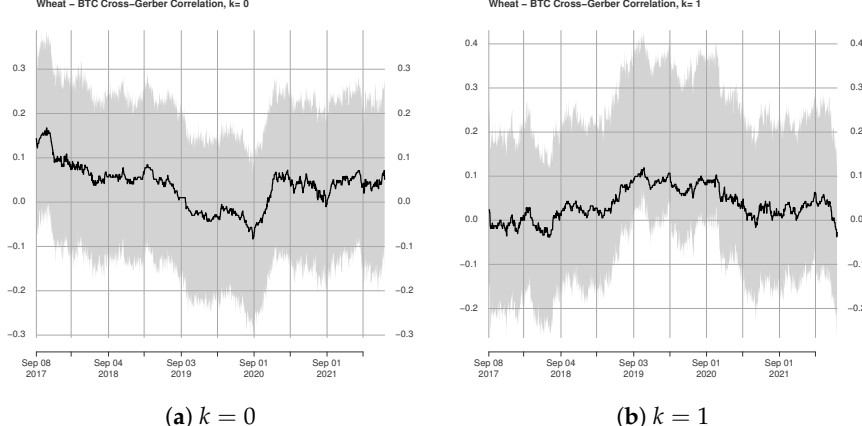

**Figure 7.** *Cont.*

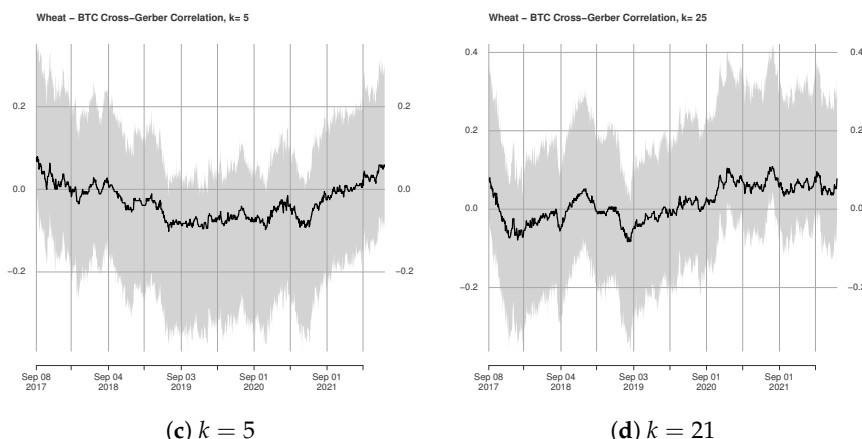

**Figure 7.** The 3-year trailing Gerber cross-correlations and 99% confidence bands, $y_1$ = Wheat, $y_2$ = BTC. Note: Thresholds are $H_i = \sigma_i/2$ where $\sigma_i$ are the (unconditional) return volatilities for $i = 1, 2$.

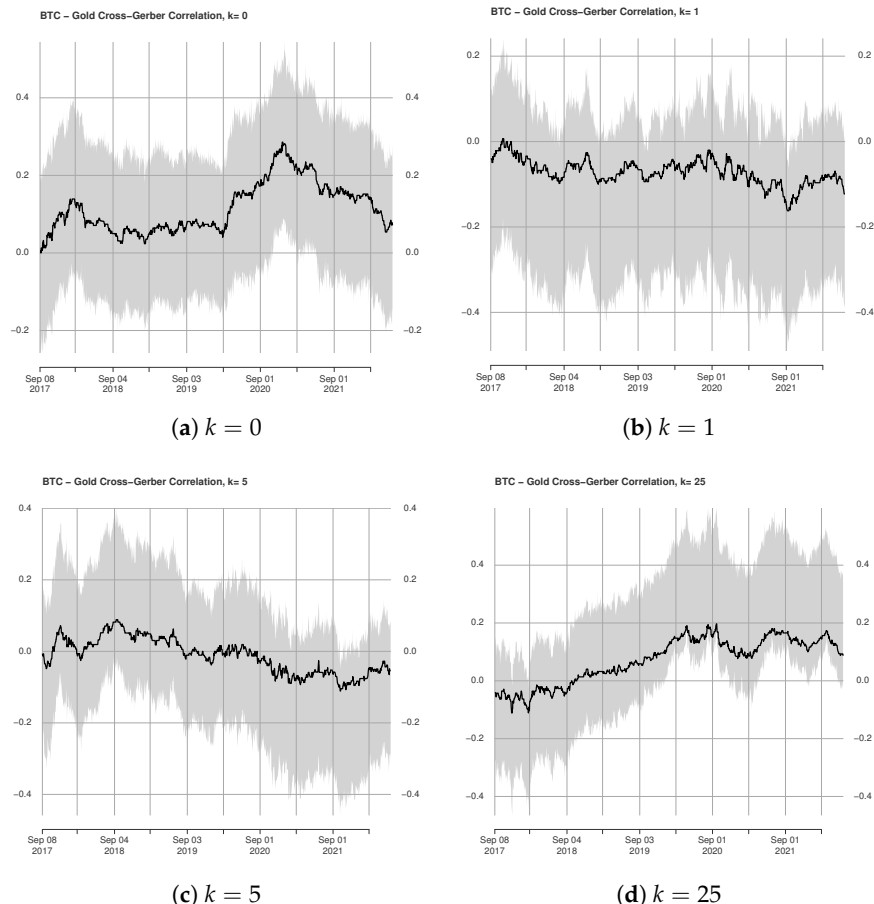

**Figure 8.** The 3-year trailing Gerber cross-correlations and 99% confidence bands, $y_1$ = BTC, $y_2$ = Gold. Note: Thresholds are $H_i = \sigma_i/2$ where $\sigma_i$ are the (unconditional) return volatilities for $i = 1, 2$.

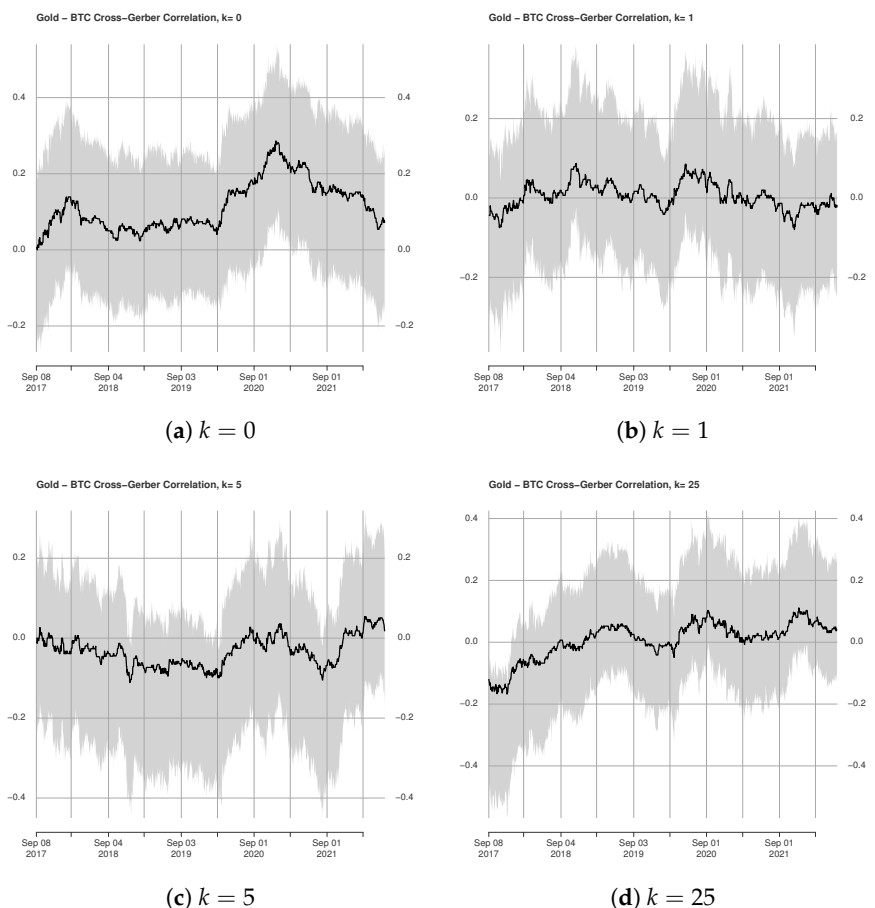

**Figure 9.** The 3-year  trailing Gerber cross-correlations and 99% confidence bands, $y_1$ = Gold, $y_2$ = BTC. Note: Thresholds are $H_i = \sigma_i/2$ where $\sigma_i$ are the (unconditional) return volatilities for $i = 1, 2$.

Finally, we test the null $H_0 : g(1) = g(2) = \ldots = g(k_{\max}) = 0$ for $k_{\max} \in \{1, 5, 10, 25\}$. Table 3 reports the values of the test statistic $\widehat{Q}(k_{\max})$ and the corresponding $p$-value $\widehat{p}(k_{\max})$. The bootstrap $p$-values are based on $B = 1000$ replicas. The only time when we reject the null is for the case $y_1$ = WTI, $y_2$ = BTC with $k_{\max} = 1$. These findings imply that there is no dependence between log returns of Bitcoin and the major categories of commodities.

**Table 3.** Bootstrap test Bitcoin, commodities.

| | $y_1$ = BTC, $y_2$ = WTI | | $y_1$ = BTC, $y_2$ = Platinum | | $y_1$ = BTC, $y_2$ = Wheat | | $y_1$ = BTC, $y_2$ = Gold | |
|---|---|---|---|---|---|---|---|---|
| $k_{\max}$ | $\widehat{Q}(k_{\max})$ | $\widehat{p}(k_{\max})$ | $\widehat{Q}(k_{\max})$ | $\widehat{p}(k_{\max})$ | $\widehat{Q}(k_{\max})$ | $\widehat{p}(k_{\max})$ | $\widehat{Q}(k_{\max})$ | $\widehat{p}(k_{\max})$ |
| 1 | 2.8590 | 0.4570 | 2.7277 | 0.5500 | 0.0083 | 0.9600 | 3.8789 | 0.5790 |
| 5 | 28.8201 | 0.4320 | 11.9621 | 0.9000 | 40.5485 | 0.7110 | 13.3090 | 0.9120 |
| 10 | 46.9309 | 0.6870 | 53.2732 | 0.9080 | 74.0836 | 0.8400 | 31.3360 | 0.9640 |
| 25 | 137.9548 | 0.9130 | 112.1169 | 0.9880 | 166.4217 | 0.9670 | 120.1614 | 0.9920 |
| | $y_1 =$ WTI, $y_2 =$ BTC | | $y_1 =$ Platinum, $y_2 =$ BTC | | $y_1 =$ Wheat, $y_2 =$ BTC | | $y_1 =$ Gold, $y_2 =$ BTC | |
| $k_{\max}$ | $\widehat{Q}(k_{\max})$ | $\widehat{p}(k_{\max})$ | $\widehat{Q}(k_{\max})$ | $\widehat{p}(k_{\max})$ | $\widehat{Q}(k_{\max})$ | $\widehat{p}(k_{\max})$ | $\widehat{Q}(k_{\max})$ | $\widehat{p}(k_{\max})$ |
| 1 | 20.2080 | 0.0410 | 8.6844 | 0.5050 | 3.4788 | 0.4300 | 0.3950 | 0.7700 |
| 5 | 43.5911 | 0.2180 | 26.5014 | 0.8040 | 15.2411 | 0.8270 | 12.5860 | 0.8540 |
| 10 | 85.0695 | 0.2350 | 44.3850 | 0.9250 | 31.5308 | 0.9300 | 28.6693 | 0.9560 |
| 25 | 108.5652 | 0.8880 | 66.4093 | 0.9980 | 97.5479 | 0.9880 | 97.3701 | 0.9930 |

We also investigate whether there is a dependence between squared Bitcoin log returns (a proxy for Bitcoin volatility) and lead/lags of log returns from the four commodities. Results are given in Table 4. Both in the case when squared Bitcoin log returns are assumed to be the $y_1$ variable and when it is assumed to be the $y_2$ variable, we do not reject the null hypothesis. This is true for all the values of $k_{max}$ we consider. Hence we can conclude that there is no dependence between Bitcoin volatility (proxied by squared Bitcoin log returns) and the returns of crude oil WTI, platinum, wheat, and gold.

**Table 4.** Bootstrap test, squared Bitcoin, commodities.

| | $y_1 = $ BTC$^2$, $y_2 = $ WTI | | $y_1 = $ BTC$^2$, $y_2 = $ Platinum | | $y_1 = $ BTC$^2$, $y_2 = $ Wheat | | $y_1 = $ BTC$^2$, $y_2 = $ Gold | |
|---|---|---|---|---|---|---|---|---|
| $k_{max}$ | $\hat{Q}(k_{max})$ | $\hat{p}(k_{max})$ | $\hat{Q}(k_{max})$ | $\hat{p}(k_{max})$ | $\hat{Q}(k_{max})$ | $\hat{p}(k_{max})$ | $\hat{Q}(k_{max})$ | $\hat{p}(k_{max})$ |
| 1 | 12.0200 | 0.5480 | 0.9370 | 0.7950 | 16.1488 | 0.3110 | 13.1282 | 0.3170 |
| 5 | 58.6704 | 0.7580 | 14.5683 | 0.9510 | 52.9512 | 0.6780 | 97.1572 | 0.3190 |
| 10 | 178.7206 | 0.7110 | 75.9803 | 0.8770 | 94.2911 | 0.8270 | 260.5584 | 0.1840 |
| 25 | 672.2332 | 0.8990 | 183.1941 | 0.9870 | 275.7607 | 0.9490 | 684.0197 | 0.2120 |
| | $y_1 = $ WTI, $y_2 = $ BTC$^2$ | | $y_1 = $ Platinum, $y_2 = $ BTC$^2$ | | $y_1 = $ Wheat, $y_2 = $ BTC$^2$ | | $y_1 = $ Gold, $y_2 = $ BTC$^2$ | |
| $k_{max}$ | $\hat{Q}(k_{max})$ | $\hat{p}(k_{max})$ | $\hat{Q}(k_{max})$ | $\hat{p}(k_{max})$ | $\hat{Q}(k_{max})$ | $\hat{p}(k_{max})$ | $\hat{Q}(k_{max})$ | $\hat{p}(k_{max})$ |
| 1 | 4.2667 | 0.6090 | 19.5400 | 0.1990 | 0.1041 | 0.9280 | 1.0248 | 0.7990 |
| 5 | 72.4407 | 0.6180 | 71.6663 | 0.4770 | 25.2471 | 0.8870 | 72.1157 | 0.4660 |
| 10 | 229.8511 | 0.3820 | 152.8596 | 0.5500 | 63.1620 | 0.9220 | 191.9757 | 0.2990 |
| 25 | 506.2262 | 0.5990 | 295.6690 | 0.9330 | 339.1286 | 0.8870 | 510.9055 | 0.3830 |

Finally, we study the dependence between squared Bitcoin log returns and lead/lags of squared log returns from the four commodities (note that when both $y_1$ and $y_2$ take on only positive values and the thresholds are positive, then $f_k^d = 0$. Therefore, in this case (3) is not suitable and we use an alternative version of the Gerber statistic discussed by [10]. Hence, only for this final case, we change the denominator of (3) and use $1 - f_k^n$ where $f_k^n = \frac{1}{T-k} \sum_{t=k+1}^{T} [I(-H_{1,t} < y_{1,t} < H_{1,t}) I(-H_{2,t-k} < y_{2,t-k} < H_{2,t-k})])$. Table 5 reports the results. From the table, it is evident that for all the levels of $k_{max}$ and for all the pairs of squared returns we consider, we strongly reject the null hypothesis, except when $y_1$ is Bitcoin and $y_2$ is WTI and vice versa. Therefore, we can conclude that there is interdependence between Bitcoin volatility and the volatility of platinum, wheat, and gold.

**Table 5.** Bootstrap test, squared Bitcoin, squared commodities.

| | $y_1 = $ BTC$^2$, $y_2 = $ WTI$^2$ | | $y_1 = $ BTC$^2$, $y_2 = $ Platinum$^2$ | | $y_1 = $ BTC$^2$, $y_2 = $ Wheat$^2$ | | $y_1 = $ BTC$^2$, $y_2 = $ Gold$^2$ | |
|---|---|---|---|---|---|---|---|---|
| $k_{max}$ | $\hat{Q}(k_{max})$ | $\hat{p}(k_{max})$ | $\hat{Q}(k_{max})$ | $\hat{p}(k_{max})$ | $\hat{Q}(k_{max})$ | $\hat{p}(k_{max})$ | $\hat{Q}(k_{max})$ | $\hat{p}(k_{max})$ |
| 1 | 1.8448 | 0.2820 | 16.9378 | 0.0000 | 9.4301 | 0.0380 | 17.7234 | 0.0000 |
| 5 | 17.9767 | 0.2060 | 60.2868 | 0.0020 | 69.5256 | 0.0000 | 74.4307 | 0.0000 |
| 10 | 31.2844 | 0.2050 | 118.3702 | 0.0000 | 137.8887 | 0.0000 | 135.3187 | 0.0000 |
| 25 | 53.1649 | 0.2070 | 308.7823 | 0.0000 | 333.0107 | 0.0000 | 316.5219 | 0.0000 |
| | $y_1 = $ WTI$^2$, $y_2 = $ BTC$^2$ | | $y_1 = $ Platinum$^2$, $y_2 = $ BTC$^2$ | | $y_1 = $ Wheat$^2$, $y_2 = $ BTC$^2$ | | $y_1 = $ Gold$^2$, $y_2 = $ BTC$^2$ | |
| $k_{max}$ | $\hat{Q}(k_{max})$ | $\hat{p}(k_{max})$ | $\hat{Q}(k_{max})$ | $\hat{p}(k_{max})$ | $\hat{Q}(k_{max})$ | $\hat{p}(k_{max})$ | $\hat{Q}(k_{max})$ | $\hat{p}(k_{max})$ |
| 1 | 3.0219 | 0.2290 | 9.8020 | 0.0430 | 9.9330 | 0.0470 | 14.4043 | 0.0000 |
| 5 | 23.6538 | 0.1600 | 71.9153 | 0.0000 | 72.8432 | 0.0010 | 73.0970 | 0.0000 |
| 10 | 44.6466 | 0.1700 | 144.1266 | 0.0000 | 138.5131 | 0.0010 | 134.3093 | 0.0000 |
| 25 | 104.3157 | 0.1820 | 326.3295 | 0.0030 | 348.7731 | 0.0020 | 308.8461 | 0.0000 |

## 4. Conclusions

The present study has explored the cross-correlation relationships between Bitcoin and the main energy, precious metal, and agricultural commodities: crude oil WTI, platinum, gold, and wheat. The analysis covers the period 2014–2022 using daily futures price data. We adopted the time-varying cross-correlation metric proposed by Gerber but readapted it to consider both contemporaneous relationships across variables and time asymmetric linkages. This means that pairs of variables have been considered at time $t$ and $t - k$. The findings of our analysis show that Bitcoin and commodities have a low cross-correlation that always stays below 0.25 over time. The low correlations between Bitcoin and other commodities might highlight that the drivers of Bitcoin's returns differ from those of other commodities. Hence it might be challenging to understand, looking at the commodity market, what could drive sudden movements in Bitcoin prices. Generally, the cross-correlations showed a tendency to rise after the COVID-19 pandemic. Furthermore, gold tends to have a weak and sometimes negative linkage with Bitcoin. Finally, there seems to be no dependence between Bitcoin volatility (proxied by squared Bitcoin log returns) and the returns of the four commodities. Instead, we found strong dependence between Bitcoin volatility and the volatility of platinum, wheat, and gold (again proxied by their squared returns).

Overall, we show that the Gerber statistic is a straightforward and robust tool, which can give insightful indications to portfolio managers building optimal strategies. For example, in our analysis, we show that while the low average correlations between Bitcoin and commodities returns might indicate the potential diversification effects for portfolios of commodities, the time-varying and volatility analysis suggest that this might not result in a reduction in risk. In fact, in times of turmoil, the correlations between Bitcoin and other commodities returns and volatility tend to increase, hurting portfolio diversification when it is needed the most.

Our analysis also has policy implications. As more and more developing economies are embarking on the usage of cryptocurrencies-based digital assets, our findings point out that these decisions should be taken with caution. Since many developing economies are commodity export dependent, linking the currency to a highly volatile asset that tends to have a higher correlation with commodities returns in times of turmoil might add additional risks to a country's stability.

The limitations of the present study are related to the fact that we only analyse two asset classes (commodities and cryptocurrencies, namely the Bitcoin). Therefore, future research involving our novel methodology could be extended to stock and bond markets, more cryptocurrencies (Ethereum, to start with) and traditional currencies.

**Author Contributions:** Conceptualisation, B.A. and A.L.; methodology, A.L.; software, K.K.L. and A.L.; validation, B.A., A.L. and L.I.; formal analysis, K.K.L.; investigation, K.K.L., B.A., A.L. and L.I.; resources, K.K.L.; data curation, B.A. and A.L.; writing—original draft preparation, K.K.L.; writing—review and editing, B.A., A.L. and L.I.; visualisation, A.L.; supervision, B.A., A.L., L.I.; project administration, K.K.L., B.A., A.L. and L.I.; funding acquisition, K.K.L. All authors have read and agreed to the published version of the manuscript.

**Funding:** This research received no external funding.

**Data Availability Statement:** 3rd Party Data. Restrictions apply to the availability of these data. Data were obtained from Bloomberg.

**Conflicts of Interest:** The authors declare no conflict of interest.

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
