# Peer review of "Exploring Dependence Relationships between Bitcoin and Commodity Returns: An Assessment Using the Gerber Cross-Correlation"

_commodities, doi:10.3390/commodities1010004_

Round 1
Reviewer 1 Report
Dear Authors
Thank you for your interesting paper.
The main novelty resides on the use of Gerber Cross-Correlation to a topic that has been extensively and increasingly studied (under a plethora of methods, as pointed by e.g. Almeida, J., & Gonçalves, T. C. (2022). A Systematic Literature Review of Volatility and Risk Management on Cryptocurrency Investment: A Methodological Point of View. Risks, 10(5), 107.). I feel there is room to contribute to the literature, but some concerns/comments require your reflection and discussion on the paper:
1- In terms of Format, please consider a thorough editing and proof reading to eliminate some typo and polish the paper (e.g. last paragraph in page 2 could benefit from a spliting into 2/3 pargraphs; line 141, p. 6 you probably do not want to say "turing" (but tuning?). I also feel that the title should clearly show that you only use BTC and not a group of cryptocurrencies. (or consider justifying that in the paper).
2- Please consider discussing some of your choices of data: why did you select the time window between 2014 and 2022, june? Why did you choose commodities future prices, rather than spot prices? Why focus your research on returns and not on volatilty? Extant literature has doccumented stronger volatility spillovers than cross correlation of returns.
3- Please explain why you test squared bitcoin log returns against first changes of log prices of the remaining commodities? Do your results hold for cross correlation of volatility proxies of both BTC and commodities?
4 - Finally, can you reconcile (or contrast) your results against those of previous literature, using different cross correlation technology/methodologies?
Kind regards
Reviewer 2 Report
The objective of the article to explore the cross-correlation relationships between Bitcoin and four commodity categories (crude oil WTI, platinum, gold and wheat) in order to determine how strongly the assets are interlinked and how they can influence each other. The purpose is interesting for people and the article is well written. It is the first article study using Gerber correlation to analyze the dependence between Bitcoin and commodities and it makes several contributions to the theory. So, the article is a good candidate for publication. However, before that, some weaknesses must be addressed:
-The authors should refine the abstract to stress the need for the present research.
-In the section named conclusions the authors must indicate: (1) which are the practical implications of the study, specifying concrete actions for decision-makers; and (2) limitations and future research (this one, related to theory extension)
